# Senecavirus A-induced glycolysis facilitates virus replication by promoting lactate production that attenuates the interaction between MAVS and RIG-I

Huizi Li[1], Cunhao Lin[1], Wenbao Qi[1], Zhenzhen Sun[1], Zhenxin Xie[1], Weixin Jia[1], Zhangyong Ning©[1,2]*

**1** College of Veterinary Medicine, South China Agricultural University, Guangzhou, China, **2** Maoming Branch, Guangdong Laboratory for Lingnan Modern Agriculture, Maoming, China

* ningzhyong@scau.edu.cn

**Data Availability Statement:** Data generated or analyzed during this study are available in the main text or the supplementary materials.

## Abstract

Senecavirus A (SVA)-induced porcine idiopathic vesicular disease has caused huge economic losses worldwide. Glucose metabolism in the host cell is essential for SVA proliferation; however, the impact of the virus on glucose metabolism in host cells and the subsequent effects are still unknown. Here, glycolysis induced by SVA is shown to facilitate virus replication by promoting lactate production, which then attenuates the interaction between the mitochondrial antiviral-signaling protein (MAVS) and retinoic acid-inducible gene I (RIG-I). SVA induces glycolysis in PK-15 cells, as indicated by significantly increased expression of hexokinase 2 (HK2), 6-phosphofructokinase (PFKM), pyruvate kinase M (PKM), phosphoglycerate kinase 1 (PGK1), hypoxia-inducible factor-1 alpha (HIF-1α), and superoxide dismutase-2 (SOD2) in a dose-and replication-dependent manner, and enhanced lactate production, while reducing ATP generation. Overexpression of PKM, PGK1, HIF-1α, and PDK3 in PK-15 cells and high glucose concentrations promote SVA replication, while glycolytic inhibitors decrease it. Inhibition of RLR signaling allowed better replication of SVA by promoting lactate production to attenuate the interaction between MAVS and RIG-I, and regulatory effect of glycolysis on replication of SVA was mainly via RIG-I signaling. SVA infection in mice increased expression of PKM and PGK1 in tissues and serum yields of lactate. Mice treated with high glucose and administered sodium lactate showed elevated lactate levels and better SVA replication, as well as suppressed induction of RIG-I, interferon beta (IFNβ), IFNα, interferon-stimulated gene 15 (ISG15), and interleukin 6 (IL-6). The inhibitory effect on interferons was lower in mice administered sodium oxamate and low glucose compared to the high glucose, indicating that RLR signaling was inhibited by SVA infection through lactate *in vitro* and *in vivo*. These results provide a new perspective on the relationship between metabolism and innate immunity of the host in SVA infection, suggesting that glycolysis or lactate may be new targets against the virus.

**Funding:** Z.N. was supported by the Start-up Research Project of Maoming Laboratory (2021TDQD002), the Natural Science Foundation of Guangdong Province (2019A1515011735) and Project of Swine Innovation Team in Guangdong Modern Agricultural Research System (2020KJ126). The funders had no role in study design, data collection, and analysis, decision to publish, or manuscript preparation.

**Competing interests:** The authors declare that they have no conflict of interest.

## Author summary

As an emerging virus, SVA induced porcine idiopathic vesicular has caused significant losses worldwide. Existing researches indicates that the ability of SVA to break through the immune barrier of the host mainly due to antagonize type I interferons (IFNs) of the host by the virus and its encoded proteins. However, persistence of the infection which made prevention and control of the disease difficulty, suggesting that other ways may exist for the virus to escape immunological surveillance by the host, and this possibility deserves further exploration. Considering that the energy generated by glucose metabolism of the host cell is needed for virus replication, we attempt to analyze the interaction of SVA and glucose metabolism of host cells and its subsequent effects. In this study, SVA infection induced glycolysis to facilitate its replication by promoting lactate production to inhibits type I IFNs mainly via RIG-I signaling by attenuating the interaction between MAVS and RIG-I was reported. This provides a new perspective on SVA infection from the innate immunity and metabolism of the host and suggests that regulation of glycolysis or targeting lactate production may be new strategies against the virus.

## Introduction

Senecavirus A (SVA) is a non-enveloped, positive-sense, single-stranded RNA virus belonging to the family *Picornaviridae* and the genus *Senecavirus* [1]. The SVA genome contains an exclusive open reading frame that encodes a polyprotein. The polyprotein is cleaved into four structural proteins (VP1, VP2, VP3, and VP4) and eight nonstructural proteins (L, 2A, 2B, 2C, 3A, 3B, 3C, and 3D) during infection and proliferation of the virus [2]. SVA causes porcine idiopathic vesicular disease, with typical clinical symptoms of vesicular lesions in the snout, oral cavities, coronary bands, and lameness. Since 2007, this disease has shown an increasing trend in major pig-farming countries, including the United States [3], Brazil [4], Colombia [5], Thailand [6], Vietnam [7], and China [8]. Notably, since 2015, when the first case occurred in China, the disease has been detected in more than 20 provinces in the country's main pork production areas [8]. The losses caused by the disease are progressively worsening worldwide, arousing great concern.

The most important characteristic of porcine idiopathic vesicular disease is the persistence of the infection, because SVA has the ability to break through the immune barrier of the host [9]. The main effect of SVA and its coded proteins is the suppression of type I IFNs that induce the antiviral responses of the host [10]. However, the difficulty of prevention and control of the disease suggests that other ways may exist for the virus to escape immunological surveillance by the host, and this possibility deserves further exploration.

Host cells and their basic metabolism are vital if viruses are to complete their life processes, including infection, replication, and release. After infection, the energy generated by host cell metabolism is needed for virus replication, as the metabolic rate determines the capacity for speed of virus replication. For this reason, viruses and their encoded proteins often create a favorable environment for virus survival and replication by inducing a series of changes in the carbohydrate metabolic pathway of the host cells. These changes include increases in glycolysis and pentose phosphate activity to support nucleotide, amino acid, and lipid synthesis [11]. The host cell, when infected by a virus, will increase its glucose intake and utilization to meet the viral requirements [12]. Glucose molecules are internalized in the host cell by glucose transporters and subsequently processed by aerobic oxidation, glycolysis, or the pentose phosphate pathway. During glycolysis, glucose is metabolized to pyruvate, which can be further metabolized in

two different ways: (1) in the presence of oxygen, pyruvate is metabolized to acetyl-CoA mainly through the mitochondrial tricarboxylic acid (TCA) cycle, which maximizes ATP production while producing minimal lactate [13]; and (2) under anaerobic conditions, the levels of key glycolytic enzymes (HK2, PFKM, PKM, and PGK1) increase, and pyruvate is instead metabolized by lactate dehydrogenase (LDH) to form large amounts of lactate [14]. Previous reports showed that many viruses, including Dengue virus (DENV) [15], Marek's disease virus (MDV) [16], White Spot Syndrome virus (WSSV) [17], Newcastle disease virus (NDV) [18], SARS-CoV-2 [19], Influenza A virus (IAV) [20], and African swine fever virus (ASFV) [21], can reprogram cellular energy metabolism to glycolysis to fulfill their replication needs. However, whether SVA infection changes the glucose metabolism of its host cells, and whether altered glucose metabolism affects the replication of the virus, is still unclear. Both viral replication and the host immune response are highly energy dependent, and research has identified some viruses that can inhibit type I IFNs by promoting glycolysis [22], however, whether the change in energy metabolism in SVA-infected cells affects the innate immune function of the host is also unclear.

The findings presented here suggest that the induction of glycolysis by SVA infection facilitated virus replication by promoting lactate production, which then attenuated the interaction between MAVS and RIG-I. Glycolysis is enhanced *in vitro* in PK-15 cells and *in vivo* in mouse tissues by SVA infection. The enhanced glycolysis promoted SVA replication, while inhibition of glycolysis suppressed virus replication. Co-immunoprecipitation (Co-IP) experiments using PK-15 cells showed that the promotion of lactate production attenuated the interaction between MAVS and RIG-I, thereby inhibiting RLR signaling and allowing better replication of SVA. Infection experiments using RIG-I-knockout (RIG-I-KO) PK-15 cells indicated that regulatory effect of glycolysis on replication of SVA was mainly via RIG-I signaling. Similar effects of SVA-induced glycolysis were verified *in vivo* in mouse experiments. Enhancement of glycolysis appears to be a new tactic used by SVA to enhance replication in the host cells, suggesting that glycolysis may also be a new target for combating the virus. These findings provide a new perspective regarding the role of host cell energy metabolism in SVA replication, as well as a better understanding of the relationship between metabolism and innate immunity in virus infections.

## Results

### Glycolysis induced by SVA infection in PK-15 and HEK293T cells

Measurement of lactate and ATP levels in PK-15 cells at 48 hours post infection (h.p.i.) with SVA at an multiplicity of infection (MOI) of 1 revealed that the lactate levels were significantly higher in SVA-infected cells than in the uninfected control cells (P < 0.01) (Fig 1A). By contrast, the ATP levels were significantly lower in the infected cells than in the control cells (P < 0.01) (Fig 1B), indicating that the virus infection promoted glycolysis in the cells.

The quantitative real-time PCR (qRT-PCR) analysis of PK-15 cells infected with Heat-SVA and SVA at MOIs of 0.1 or 1 for 48 h showed that SVA infection and replication induced dose-dependent increases in the expression of HK2, PFKM, PKM, PGK1, HIF-1α, and SOD2, but not in Heat-SVA-treated or uninfected cells (Fig 1C). Western blots of PGK1, PKM, and VP2 protein showed that SVA infection significantly increased the expression of PGK1 and PKM (P < 0.05) (Fig 1D and 1E), indicating that the induction of glycolysis by SVA infection in PK-15 cells depends on virus replication. In addition, similar effect induced by SVA infection was also found in HEK293T cells (S1 Fig).

### Enhanced glycolysis promotes replication of SVA

Transfection of pEGFP-PGK1, PKM, HIF-1α, or PDK3 into PK-15 cells revealed promotion of the mRNA expression of the SVA VP2 gene by PDK3 overexpression at 6 (P < 0.05), 12,

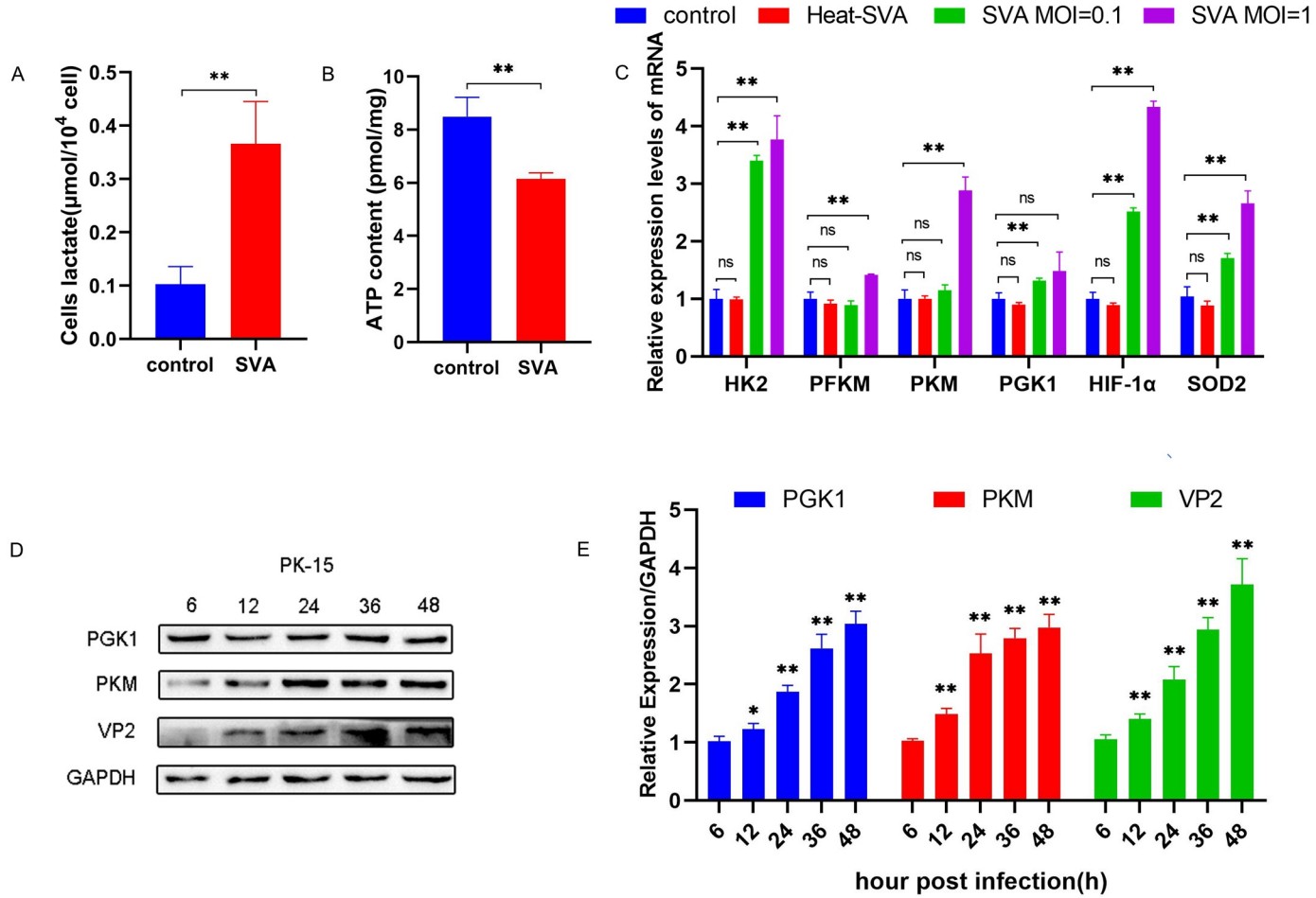

**Fig 1. SVA infection induces glycolysis in PK-15 cells. (A, B)** PK-15 cells were infected with SVA at an MOI of 1. Cell lysates were collected at 48 h.p.i. and the lactate and intracellular ATP levels were measured. **(C)** PK-15 cells were mock infected or infected with Heat-SVA and SVA at MOIs of 0.1 or 1. Cells were harvested at 48 h.p. i. The mRNA expression of HK2, PFKM, PGK1, PKM, HIF-1α, and SOD2 was analyzed by qRT-PCR. **(D)** Cells were infected with SVA at an MOI of 1 and lysed in RIPA buffer at 48 h.p.i. The expression levels of PGK1, PKM, and VP2 were analyzed by western blot. **(E)** The grayscale analysis of PGK1, PKM, and VP2 protein. All data represent the means ± SD (Student's t test) (*P < 0.05, **P < 0.01, ns, not significant).

and 24 h (P < 0.01) (Fig 2 A), by PGK1 overexpression at 6 (P < 0.01), 12 (P < 0.05), and 24 h (P < 0.01) (Fig 2B), by PKM overexpression at 6 and 24 h (P < 0.01) (Fig 2C), and by HIF-1α overexpression at 12 and 24 h compared with the uninfected control group (P < 0.01) (Fig 2D). Co-transfection with all four vectors at a 1:1:1:1 ratio to simulate the effect of pro-glycolysis conditions also promoted viral replication at 6, 12 (P < 0.01), and 24 h (P < 0.05) (Fig 2E). Measurements of SVA proliferation at different time points in PK-15 cells overexpressing PGK1, PKM, PDK3, or HIF-1α (Fig 2F) revealed significant increases in the virus titer from 6 to 24 h in PGK1-overexpressing, PDK3-overexpressing, and co-transfected cells (P < 0.01), at 6 and 24 h in PKM-overexpressing cells (P < 0.01), and at 24 h in HIF-1α-overexpressing cells (P < 0.01) compared with the infected control cells. Taken together, these results suggested that promotion of glycolysis facilitated viral replication in PK-15 cells.

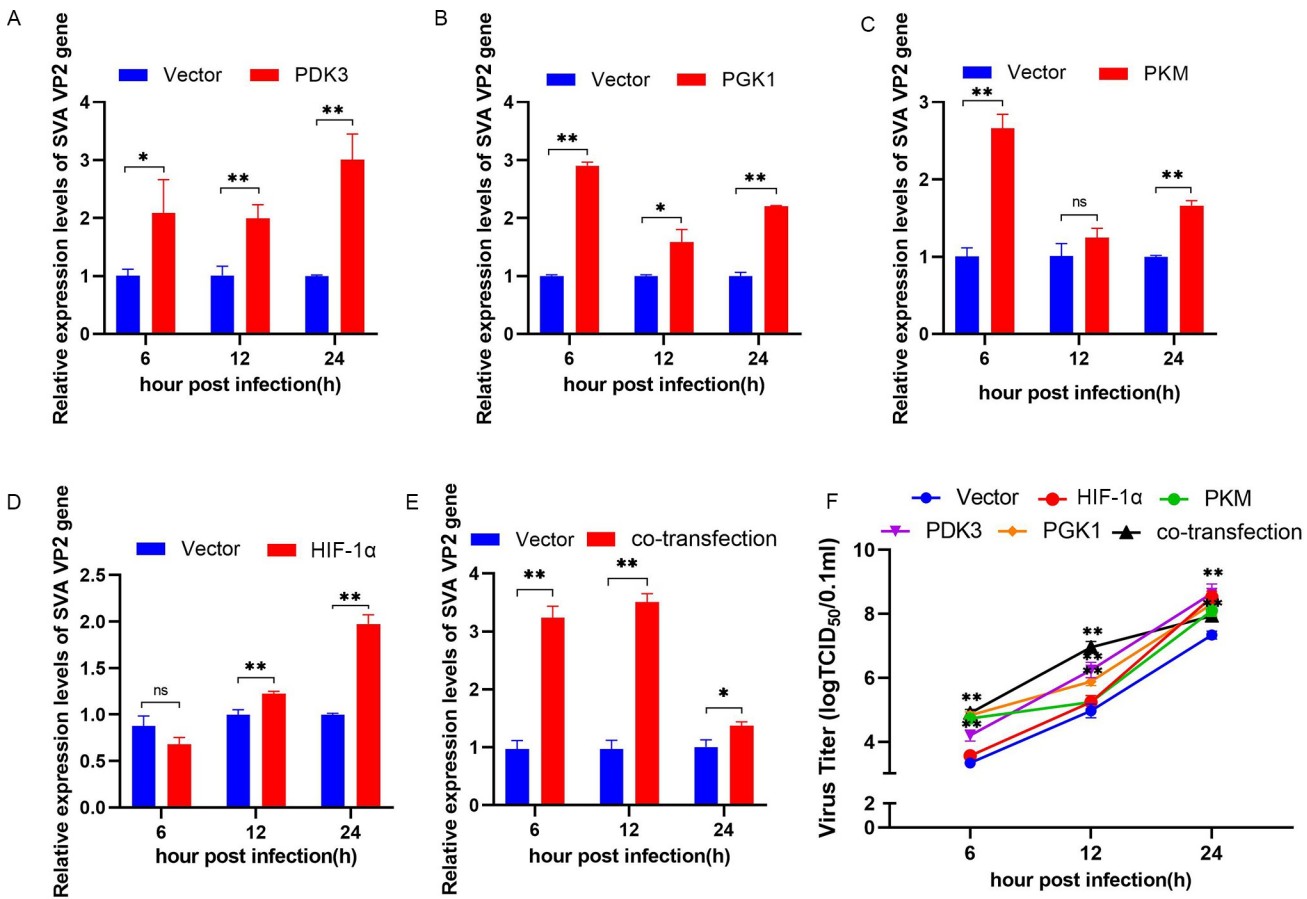

**Fig 2. Enhanced glycolysis promotes replication of SVA.** PK-15 cells were transfected with pEGFP-PDK3, pEGFP-PGK1, pEGFP-PKM, pEGFP-HIF-1α or co-transfected with all four vectors. Cells were infected with SVA at an MOI of 1. **(A-E)** The RNA of the cells was harvested and extracted at 6, 12 or 24 h.p.i, and intracellular mRNA levels of the SVA VP2 gene were analyzed by qRT-PCR. β-actin expression was used as an internal control. **(F)** Virus titers in PK-15 cells were detected using the Reed–Muench method. All data represent the means ± SD (Student's t test) (*P < 0.05, **P < 0.01, ns, not significant).

## Inhibition of glycolysis inhibits replication of SVA

Glycolysis was inhibited in PK-15 cells using three inhibitors (Fig 3A): 2DG [23], oxamate [24], and DCA [25]. PK-15 cells were infected with SVA at an MOI of 1, and then incubated in the presence or absence of inhibitor treatment for 48 h. The mRNA levels of the SVA VP2 gene were significantly decreased by treatment with 2DG (P < 0.01), oxamate (P < 0.01), and DCA (P < 0.01) (Fig 3B). The 2DG, oxamate, and DCA treatments also reduced the expression of the VP2 protein (P < 0.05) (Fig 3C and 3D).

Exposure of SVA-infected PK-15 cells (MOI of 1) to different glucose concentrations (high: 25 mM or low: 5 mM) at 48 h.p.i. revealed with high glucose significantly higher mRNA expression of the VP2 gene in cells exposed to high glucose (equivalent to increased glycolysis) than to low glucose (P < 0.01) (Fig 3E). The western blot results revealed greater expression of the viral VP2 protein in cells given the high-glucose treatment than in cells given the low glucose treatment (P < 0.01) (Fig 3F and 3G). These results indicated that glucose metabolism affected SVA replication and was related to glycolysis.

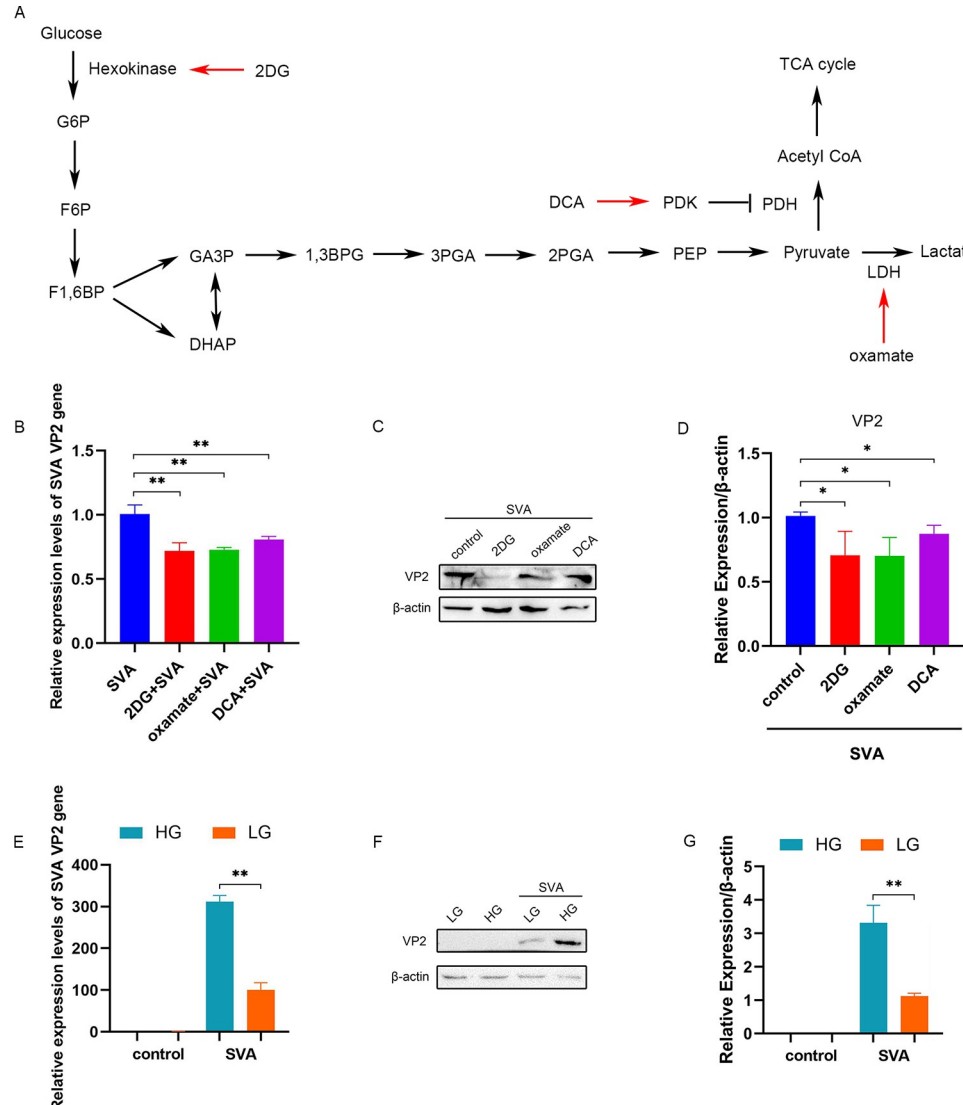

**Fig 3. Inhibition of glycolysis inhibits replication of SVA. (A)** Schematic overview of glucose metabolism and the functional targets of glycolytic inhibitors (2DG, oxamate, and DCA) used in this study. **(B)** PK-15 cells were infected with SVA at an MOI of 1, incubated in the presence or absence of glycolytic inhibitors, and intracellular mRNA levels of the SVA VP2 gene were analyzed by qRT-PCR at 48 h.p.i. **(C)** Cells were infected with SVA at an MOI of 1, incubated in the presence or absence of glycolytic inhibitors, and lysed with RIPA at 48 h.p.i. The VP2 protein levels were detected by western blot. **(D)** The grayscale analysis of the VP2 protein. **(E)** PK-15 cells were infected with SVA at an MOI of 1, incubated in high (25 mM) or low (5 mM) glucose, and intracellular mRNA levels of SVA VP2 gene were measured by qRT-PCR. **(F)** Cells were infected with SVA at an MOI of 1, incubated in high (25 mM) or low (5 mM) glucose and lysed with RIPA at 48 h.p.i. VP2 protein levels were determined by western blot. **(G)** The grayscale analysis of VP2 protein. All data represent the means ± SD (Student's t test) (*P < 0.05, **P < 0.01, ns, not significant).

## Glycolysis induced by SVA infection inhibits RLR signaling

RLR signaling is an important innate immune pathway with antiviral effects associated with the production of type I IFNs [26]. The qRT-PCR analysis of PK-15 cells infected with SVA at an MOI of 1, incubated in the presence or absence of glycolysis inhibitors revealed altered mRNA expression of IFNs and interferon-stimulating genes (ISGs) at 48 h.p.i. Specifically, the expression of RIG-I was significantly increased in the oxamate- and DCA-treated cells

(P < 0.01) (Fig 4A). IFNβ, IFNα, and IFIT1 expression was significantly increased by all three inhibitors (P < 0.01) (Fig 4B–4D). IL-6 expression was significantly increased by 2DG and oxamate (P < 0.05) (Fig 4E). ISG-15 was significantly increased by 2DG (P < 0.05), oxamate, and DCA (P < 0.01) (Fig 4F). These findings indicated a modulation of RIG-I-induced IFNs by SVA-induced glycolysis. In addition, the expression of RIG-I, IFNβ, IFNα, IL-6, IFIT1, and ISG-15 was significantly lower in cells treated with high glucose than with low glucose (P < 0.01) (Fig 4G–4L).

## Lactate produced by SVA-induced glycolysis inhibits RLR signaling

The qRT-PCR analysis of PK-15 cells with exogenous lactate, the terminal product of glycolysis and a known inhibitor of the type I interferon signaling pathway [27], revealed a significant promotion of the mRNA levels of the SVA VP2 gene. Treatment of SVA-infected PK-15 cells with increasing concentrations of lactate showed a dose and replication dependency of this response (Fig 5A). By contrast, oxamate treatment significantly suppressed the mRNA expression of the SVA VP2 gene, again in a dose- and replication-dependent manner (Fig 5B), while also significantly inhibiting lactate production (P<0.01) (Fig 5C). The VP2 protein levels were significantly increased by lactate and showed a similar dose dependency (Fig 5D), consistent with the mRNA results for SVA VP2.

Oxamate treatment eliminated the promoting effect of lactate on viral proliferation (Fig 5E and 5F). The qRT-PCR results for PK-15 cells incubated in the presence or absence of oxamate or lactate for 48 h revealed that oxamate, as a specific LDHA inhibitor, reduced the lactate levels produced by glycolysis and increased the mRNA expression of type I IFNs (RIG-I, IFNβ, IFNα, ISG-15, IL-6, and IFIT1), whereas lactate treatment increased the mRNA expression of the SVA VP2 gene and reduced the expression of IFNβ, IFNα, ISG-15, IL-6, and IFIT1 (Fig 5G–5L). These results showed that lactate treatment inhibited the expression of IFNβ and that lactate levels were negatively correlated with the RLR signaling pathway.

## Lactate inhibits RLR signaling by destabilizing the interaction between MAVS and RIG-I

Lactate negatively regulates RLR activation by targeting MAVS [28]. Co-IP experiments showed that oxamate had no effect on the interaction between RIG-I and MAVS in the absence of infection, and that oxamate suppresses the inhibitory effect of SVA on the binding of the MAVS-RIG-I complex. Oxamate treatment eliminated the effect of SVA on the interaction between RIG-I and MAVS (Fig 6A), whereas lactate treatment eliminated the positive effect of oxamate on the interaction between MAVS and RIG-I (Fig 6B), suggesting that the promotion of lactate production by SVA infection inhibits RLR signaling by destabilizing the interaction between MAVS and RIG-I.

To verify the relationship between lactate and RIG-I-mediated type I interferon pathway, prepared RIG-I-KO PK-15 cells were used for subsequent experiments (Fig 6C). There was no significant difference in IFNβ, ISG15, IFNα, and IFIT1 between SVA infected oxamate-RIG-I-KO and RIG-I-KO groups (Fig 6D–6G). Lactate has a slight inhibitory effect on ISG-15, IFNα, and IFIT1 in RIG-I-KO PK-15 cells, however, there was no significant difference compared with the SVA infected RIG-I-KO cells (Fig 6E–6G), indicating that the regulation of IFNs by lactate was mainly through RIG-I.

## RLR signaling is inhibited by lactate in SVA-infected mice

A mouse model for SVA infection [29], established by oral challenge with SVA, was used for *in vivo* verification of the *in vitro* PK-15 cell results (Fig 7A). At 7 days post infection (d.p.i.), the

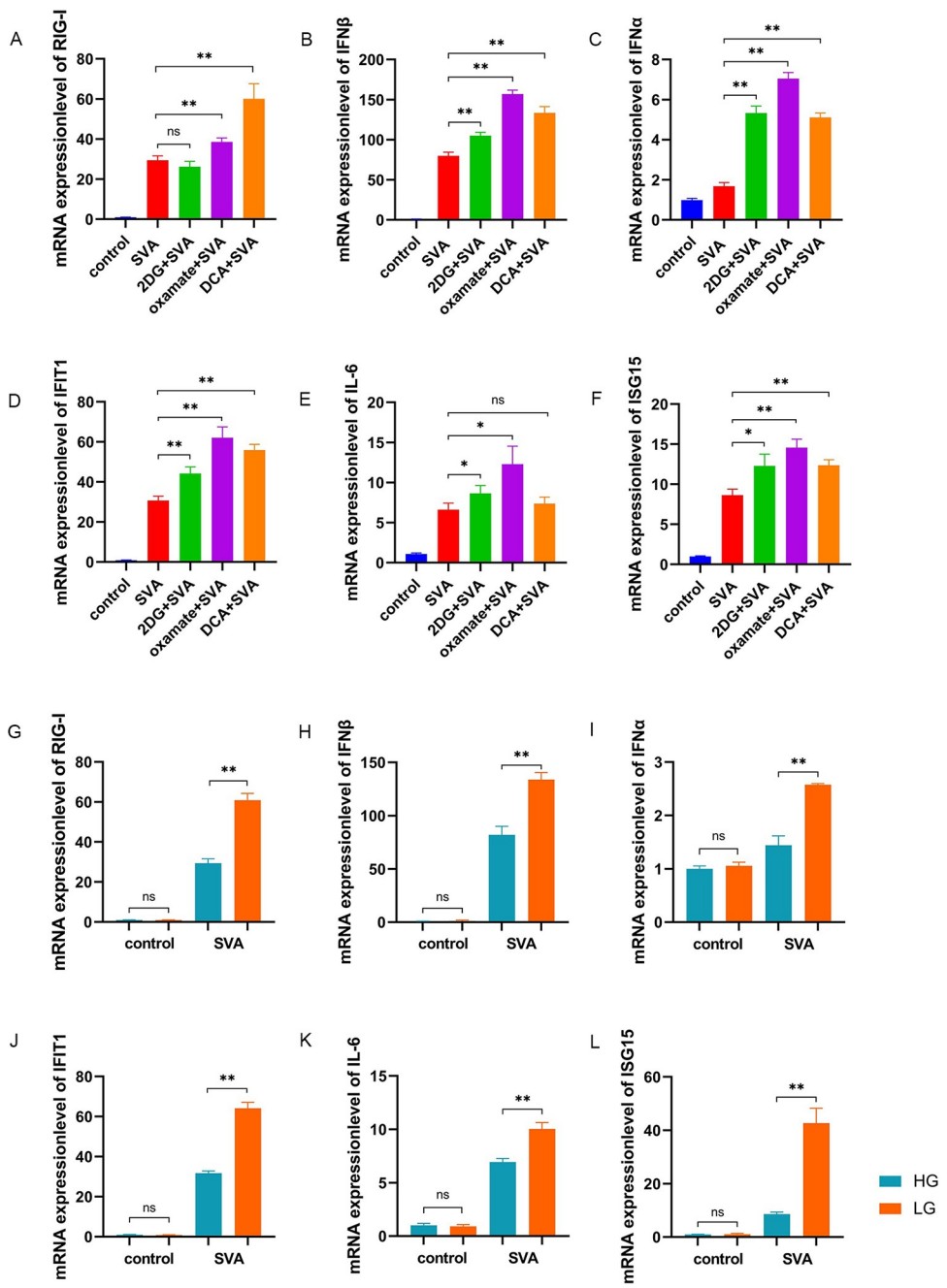

**Fig 4. Glycolysis induced by SVA infection inhibits RLR signaling. (A-F)** PK-15 cells were mock infected or infected with SVA at an MOI of 1 and incubated in the presence or absence of 2DG, DCA, or oxamate. Cells were harvested at 48 h.p.i. The mRNA expression of RIG-I, IFNβ, IFNα, IFIT1, IL-6, and ISG-15 were analyzed by qRT-PCR. **(G-L)** PK-15 cells were treated with high (25 mM) or low (5 mM) glucose and infected with SVA at an MOI of 1. Cells were harvested at 48 h.p.i. The mRNA expression of RIG-I, IFNβ, IFNα, IFIT1, IL-6, and ISG-15 were analyzed by qRT-PCR. β-actin expression was used as an internal control. All data represent the means ± SD (Student's t test) (*P < 0.05, **P < 0.01, ns, not significant).

mRNA levels of the VP2 gene were significantly higher in the heart, liver, spleen, duodenum, and kidney tissues of the SVA model mice than in the uninfected control mice, with the highest levels in the heart tissue (P < 0.01) (Fig 7B). SVA infection significantly increased the

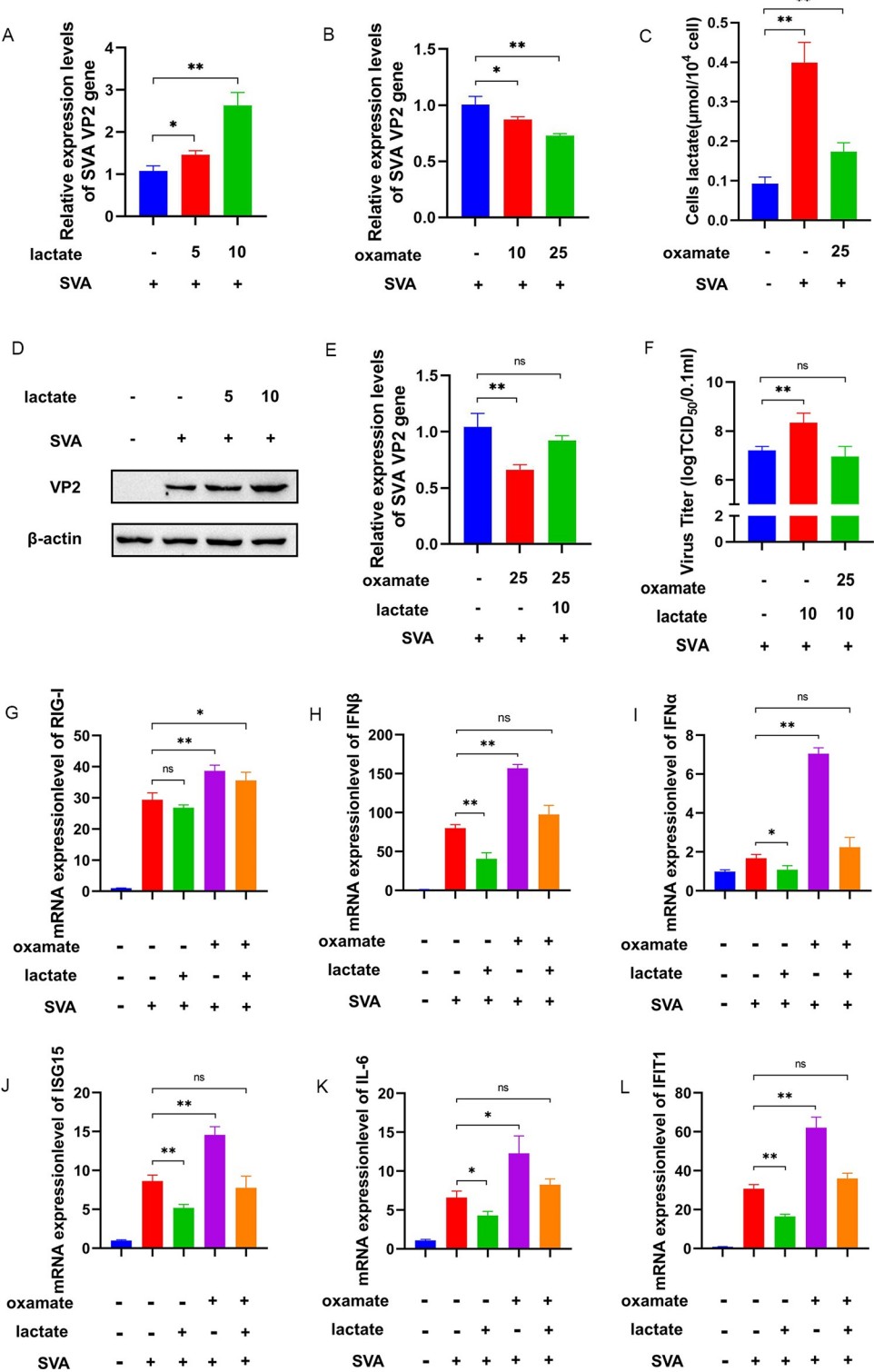

**Fig 5. Lactate produced by SVA-induced glycolysis inhibits RLR signaling. (A-F)** PK-15 cells were incubated in the presence or absence of sodium oxamate (10 or 25 mM), in the presence or absence of sodium lactate (5 or 10 mM), and with or without SVA infection at an MOI of 1. **(A, B)** The expression levels of the SVA VP2 gene were analyzed by qRT-PCR at 48 h.p.i. **(C)** The intracellular lactate levels were measured using lactate assay kits. **(D)** VP2 protein levels were measured by western blot. **(E)** The expression levels of the SVA VP2 gene were analyzed by qRT-PCR at 48 h.p.i. **(F)** Virus titers in PK-15 cells were detected using the Reed–Muench method. **(G-L)** The mRNA expression of RIG-I,

IFNβ, IFNα, ISG-15, IL-6, and IFIT1 was analyzed by qRT-PCR. β-actin expression was used as an internal control. All data represent the means ± SD (Student's t test) (*P < 0.05, **P < 0.01, ns, not significant).

production of lactate in the serum (P < 0.01) (Fig 7C) and promoted the protein expression of HIF-1α (P < 0.01), PGK1 (P < 0.01), and PKM (P < 0.01) in the spleen and PKM in the heart (P < 0.01) and liver (P < 0.05) (Fig 7D and 7E), indicating that glycolysis was induced *in vivo* by SVA infection in mice. Mice treated with high glucose showed significantly increased SVA VP2 gene expression levels (P < 0.01) (Fig 7F) and suppressed mRNA expression of IFNβ, IL-6, ISG15, RIG-I, and IFNα, compared with the SVA-challenged group (P < 0.01) (Fig 7G–7K). Low glucose conditions resulted in significantly suppressed mRNA levels for IFNβ (P < 0.05) (Fig 7G), IL-6 (P < 0.05) (Fig 7H), and ISG15 (P < 0.05) (Fig 7I), but the levels were still higher than those from mice given the high glucose treatment. No significant difference was noted in the mRNA expression of RIG-I (Fig 7J) or IFNα (Fig 7K) between the low glucose-treated and SVA-challenged groups.

Mice administered sodium oxamate showed significantly lower VP2 expression (P < 0.05) (Fig 7L) and significantly higher expression of IFNα (P < 0.01) (Fig 7M), IFNβ (P < 0.01) (Fig 7N), ISG15 (P < 0.01) (Fig 7O), RIG-I (P < 0.05) (Fig 7P), and IL-6 (P < 0.01) (Fig 7Q). By contrast, sodium lactate supplementation significantly increased the SVA mRNA levels in the heart (P < 0.01) (Fig 7L), while also significantly reducing the mRNA levels of IFNα, IFNβ, ISG15, RIG-I, and IL-6 (P < 0.01) (Fig 7M–7Q). No significant difference was observed in the expression levels of the SVA VP2 gene in the hearts of mice cotreated with both oxamate and lactate and those in the SVA-challenged group (Fig 7L), indicating that oxamate inhibited the promoting effect of lactate on SVA replication. No significant difference was detected in the expression levels of IFNα (Fig 7M), IFNβ (Fig 7N), or ISG15 (Fig 7O), whereas the expression levels of RIG-I (Fig 7P) and IL-6 (Fig 7Q) were significantly reduced (P < 0.05) compared with SVA-challenged group. However, the expression levels were still higher than those in the lactate-treated group, in agreement with the *in vitro* results for PK-15 cells, indicating that SVA inhibited RLR signaling through lactate in both the *in vivo* and *in vitro*.

The results of histopathological detection showed that mice treated by high glucose and lactate promote the damage of SVA infection with the degeneration and necrosis of mucosal epithelial cells and shedding of intestinal villi, vacuolar degeneration of renal tubular epithelial cells, and the decrease of lymphocytes in splenic corpuscle and white pulp. In addition, lactate increased the vacuolar degeneration and necrosis with fewer in quantity of liver cells. Histopathological changes were reduced in oxamate and oxamate + lactate group compared with the high glucose and lactate group. And, significant difference in pathological lesions was not observed in groups infected by SVA (Fig 8).

## Discussion

As an emerging virus, SVA has a worldwide distribution and causes huge losses in the swine industry. It causes porcine idiopathic vesicular disease, which is characterized by erosion and vesicles on the snout and in the oral cavity. It was first detected in the United States [3] and Canada [30] in 2007, but disease outbreaks have been reported since then in Brazil [4], China [31], Thailand [6], and Columbia [5]. The first outbreak in China occurred in Guangdong Province in 2015 and has been reported in the main swine-producing areas of China [8]. Asymptomatic SVA infection has also been reported in China, suggesting that the virus is a persistent infection [32] and making its prevention and control an important challenge. This further illustrates the need for in-depth research on the interactions between the virus and the immunity of its host.

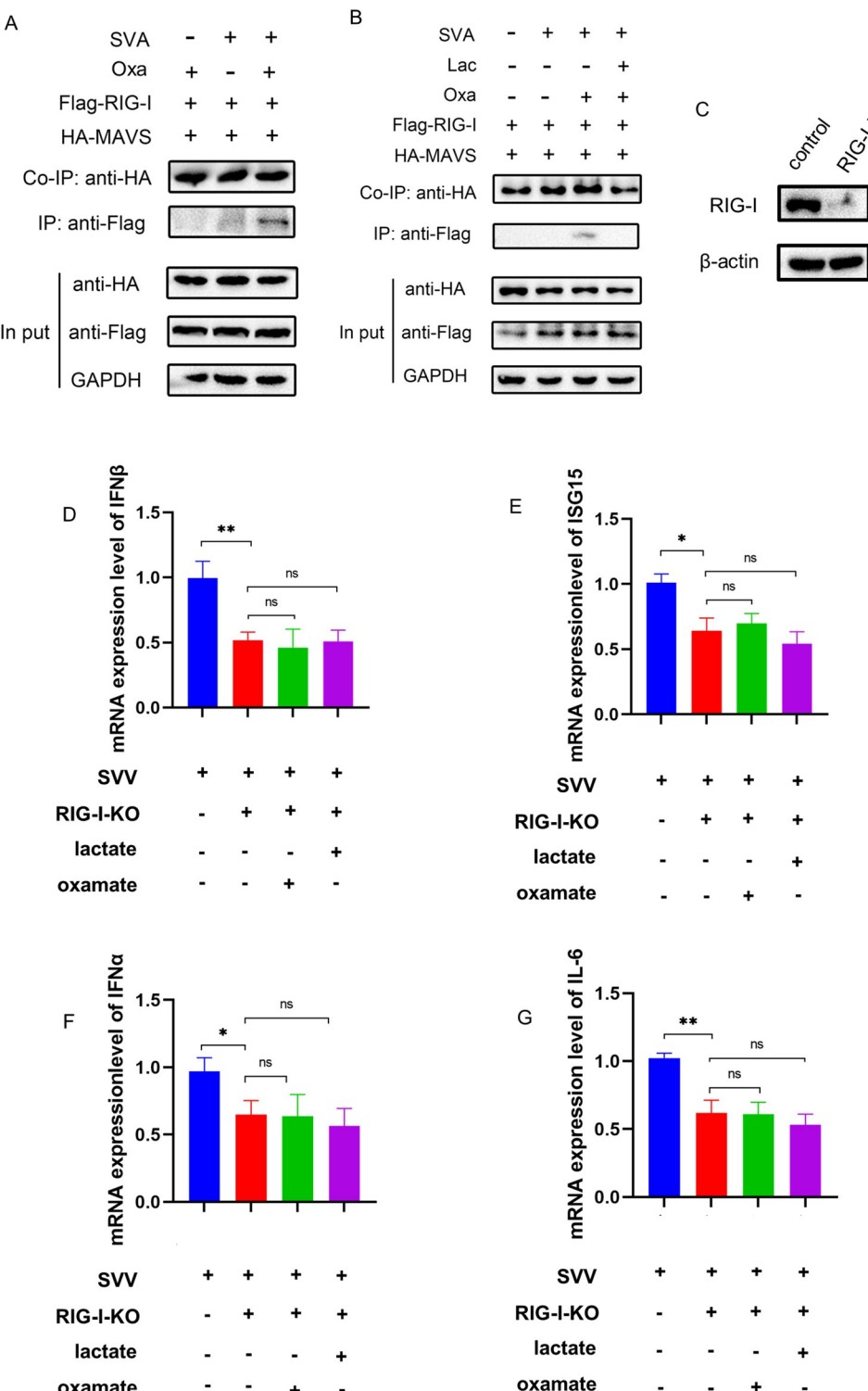

**Fig 6. Lactate produced by SVA infection inhibits RLR signaling mainly through RIG-I. (A, B)** Lactate inhibits RLR signaling by destabilizing the interaction between MAVS and RIG-I. **(C)** Expression levels of RIG-I in PK-15 cells and RIG-I-KO PK-15 cells. **(D-F)** The mRNA expression of IFNβ, ISG15, IFNα, and IL-6 was analyzed by qRT-PCR. All data represent the means ± SD (Student's t test) (*P < 0.05, **P < 0.01, ns, not significant).

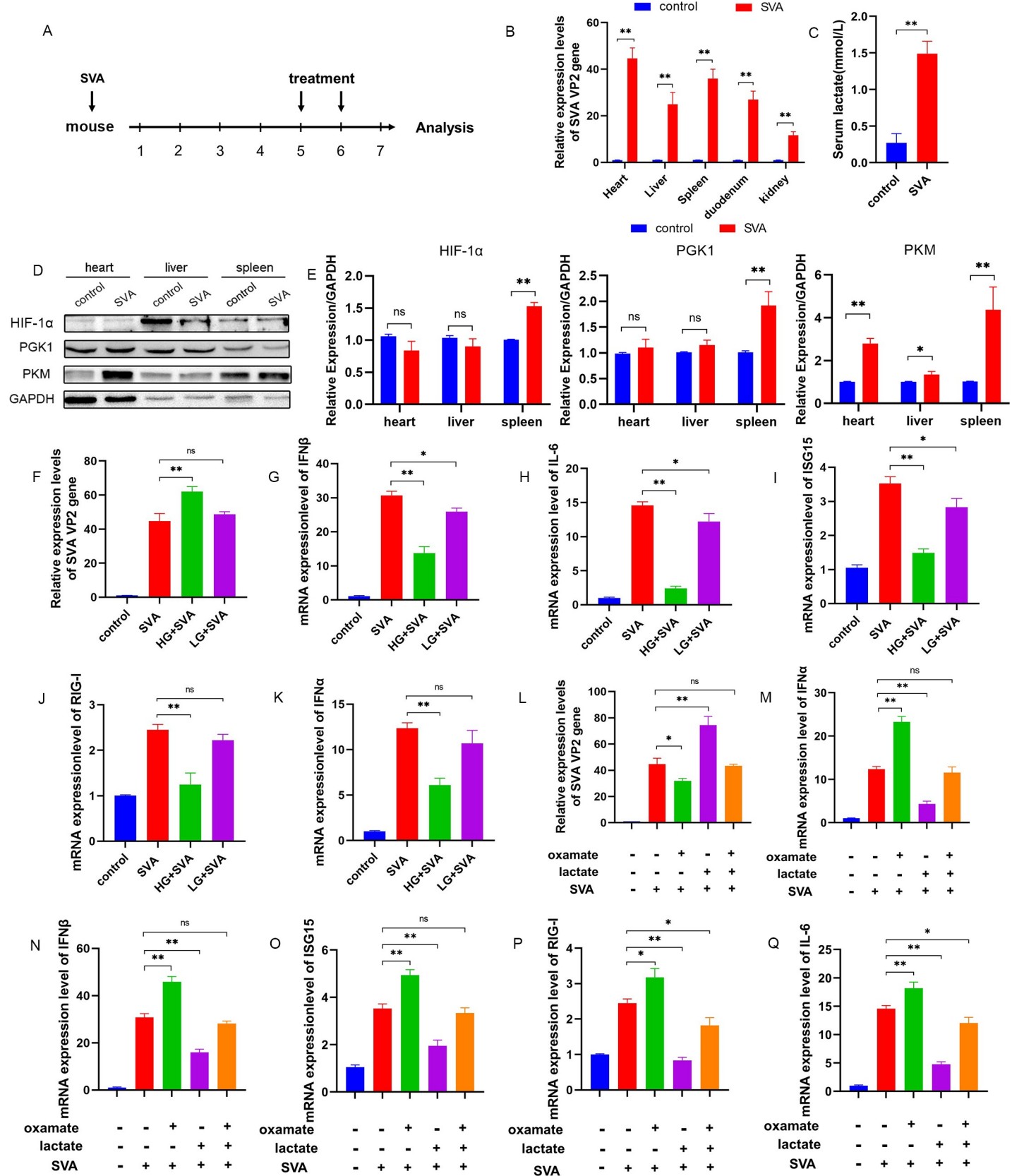

**Fig 7. RLR signaling was inhibited by SVA through lactate in mice. (A)** Mice were orally challenged with SVA, treated with high (1.5 g/kg), low (0.2 g/kg), or no (control) glucose, and treated with or without sodium oxamate (750 mg/kg) or sodium lactate (1 g/kg). **(B)** At 7 d.p.i, the expression of the SVA VP2 gene in the heart, liver, spleen, duodenum, and kidney was analyzed by qRT-PCR. **(C)** At 7 d.p.i, the serum lactate was measured using lactate assay kits. **(D)** The expression of PGK1, PKM, and HIF-1α in mouse heart, liver, and spleen tissues was detected by western blot. **(E)** Grayscale analysis of detected proteins. **(F-Q)** The expression of the SVA VP2 gene, IFNβ, IFNα, ISG15, RIG-I, and IL-6 was analyzed by qRT-PCR. All data represent the means ± SD (Student's t test) (*P < 0.05, **P < 0.01, ns, not significant).

Previous research on SVA and host immunity mainly focused on the regulation of the type I interferon signaling pathway [33]. Transcriptome analysis of SVA-infected cells showed that type I IFNs were key antiviral factors. The encoded SVA proteins also play an important role

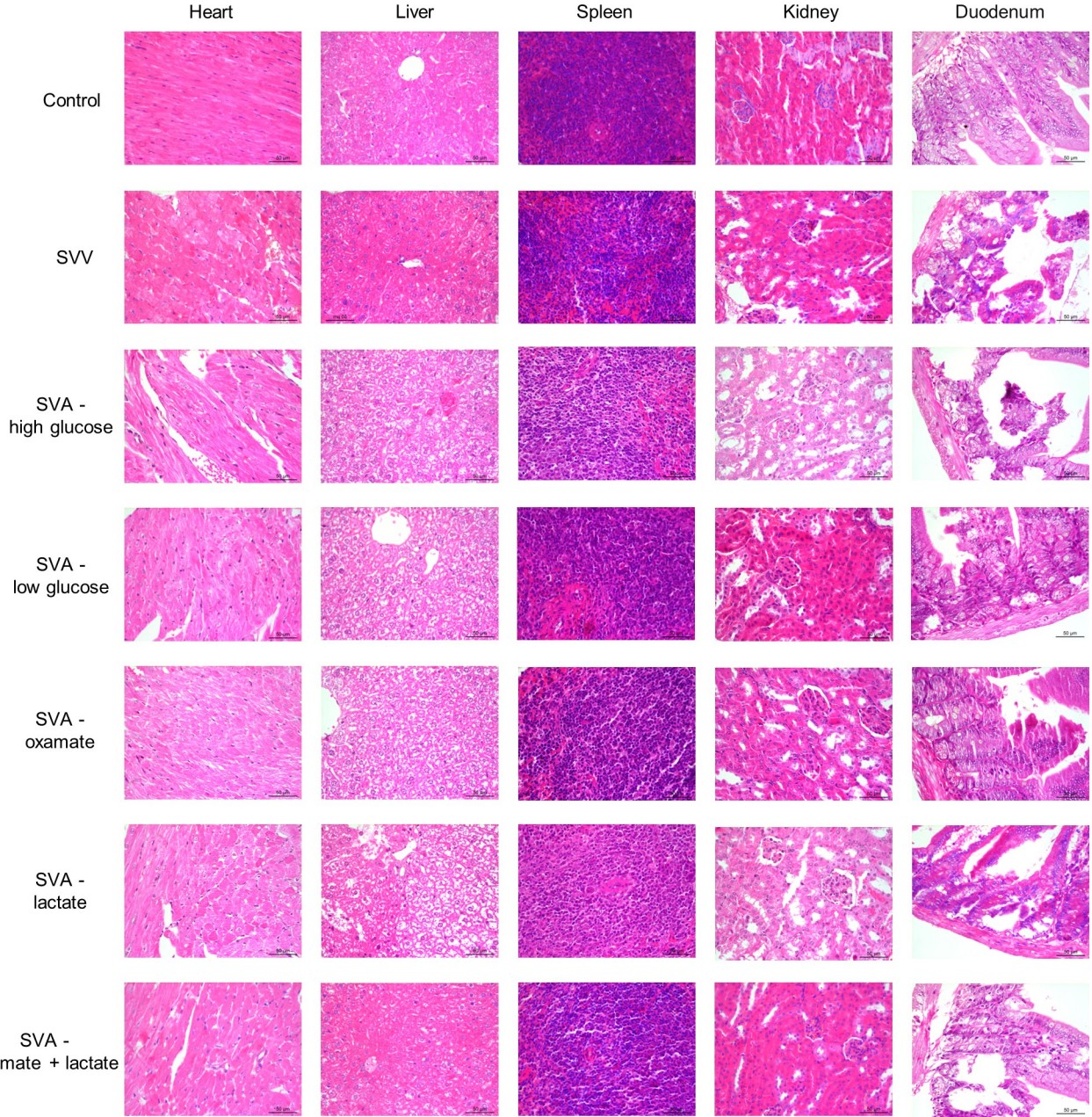

**Fig 8. Histopathological changes of heart, liver, spleen, kidney and duodenum in mice.**

in antagonizing the innate immune response dominated by IFNs. Viral 2B and 3C[pro] suppress type I IFNs production by inducing the degradation of MAVS [34], while 2C[hel] and 3C[pro] inhibit the production of RIG-I-mediated IFN-β by degrading RIG-I [10]. In addition, 3C[pro] also degrades the interferon regulatory factors IRF3 and IRF7 [35], while 3D[pol] interacts with NLRP3 to induce IL-1β production by activating NF-κB and ion channel signaling [36]. Although progress has been made in suppressing the type I interferon signaling pathway induced by SVA and its coded proteins, the possibility that the virus inhibits the innate immune system in other ways still needs to be explored.

Viral replication relies on the host metabolic network, and a virus actively reprograms metabolism to establish an optimal environment for its infection and replication. Some viruses, such as H1N1 influenza virus and MDV, have been reported to induce host glycolysis [16, 37]. A recently reported bioinformatics analysis of metabolism in SVA infection showed that the virus can reprogram cellular glucose metabolism [38], in agreement with the experimental results presented here. CH-GDFS-2018 is a wild type virus isolated by our group in 2018 with recombinant features [39]. In order to verify whether the clinically isolated strains have the same characteristics that promote glycolysis, the mRNA levels of HIF-1α, PGK1, HK2, SOD2, PKM and PFKM after cell infected with CH-GDQC-2017 [40] and CH-GDZS-2019 [41] were quantified, and the results showed that these strains had similar abilities to cause glycolysis (S2 Fig), indicating that glycolysis induced by SVA infection is not unique to a particular strain. Glycolysis is closely related to immunity, and most cancer cells favor glycolysis for energy metabolism, known as the Warburg effect, as well as high lactate production, to enhance their survival and proliferation [42]. Lactate has often been viewed as a useless metabolic end product, but some recent studies have shown that lactate plays an important role in several cellular processes, including energy regulation, immune tolerance, memory formation, wound healing, and cancer growth and metastasis [12,43]. Previous studies have shown that lactate inhibits the RLR signal transduction pathway by directly binding to MAVS, thereby reducing the production of IFNs. The present study is the first to report that SVA infection can reprogram host glucose metabolism to favor glycolysis and lactate production. This switch facilitates virus replication by attenuating the interaction between MAVS and RIG-I, thereby inhibiting the production of type I IFNs (Fig 9). These findings are consistent with the manipulation of metabolism and immunity reported for other viruses, such as hepatitis B and SARS-CoV-2 [19, 27]. The knockout of RIG-I weakened the regulatory effect of oxamate and lactate on IFNs production and lactate slightly inhibit the production of IFNs in RIG-I-KO cells, suggesting that lactate may inhibit IFNs by other pathways, however, it mainly via RIG-I signaling to manipulate SVA-induced innate immunity. The results presented here suggest that targeting key enzymes in glycolysis and its related biological processes may be an important approach for development of agents for combating SVA infection.

High glucose conditions promote the proliferation of SVA, which is consistent with findings reported for Dengue virus [44] and with the discovery that controlling high blood sugar also reduces the progression of COVID-19 to severe illness or death [45]. This suggests that blood sugar levels could be an effective measurement parameter for the pathogenicity of some viruses. In the present study, high glucose levels also inhibited the production of type I IFNs during SVA infection, both *in vivo* and *in vitro*. The results of histopathological detection in animal experiments also support this. Therefore, host glucose metabolism may be an important factor that provides a favorable environment for SVA replication while also having a close relationship with the functioning of the immune system.

The results presented here provide a new perspective on SVA infection and the relationship between metabolism and the innate immunity of the host. Glycolysis is a new pathway

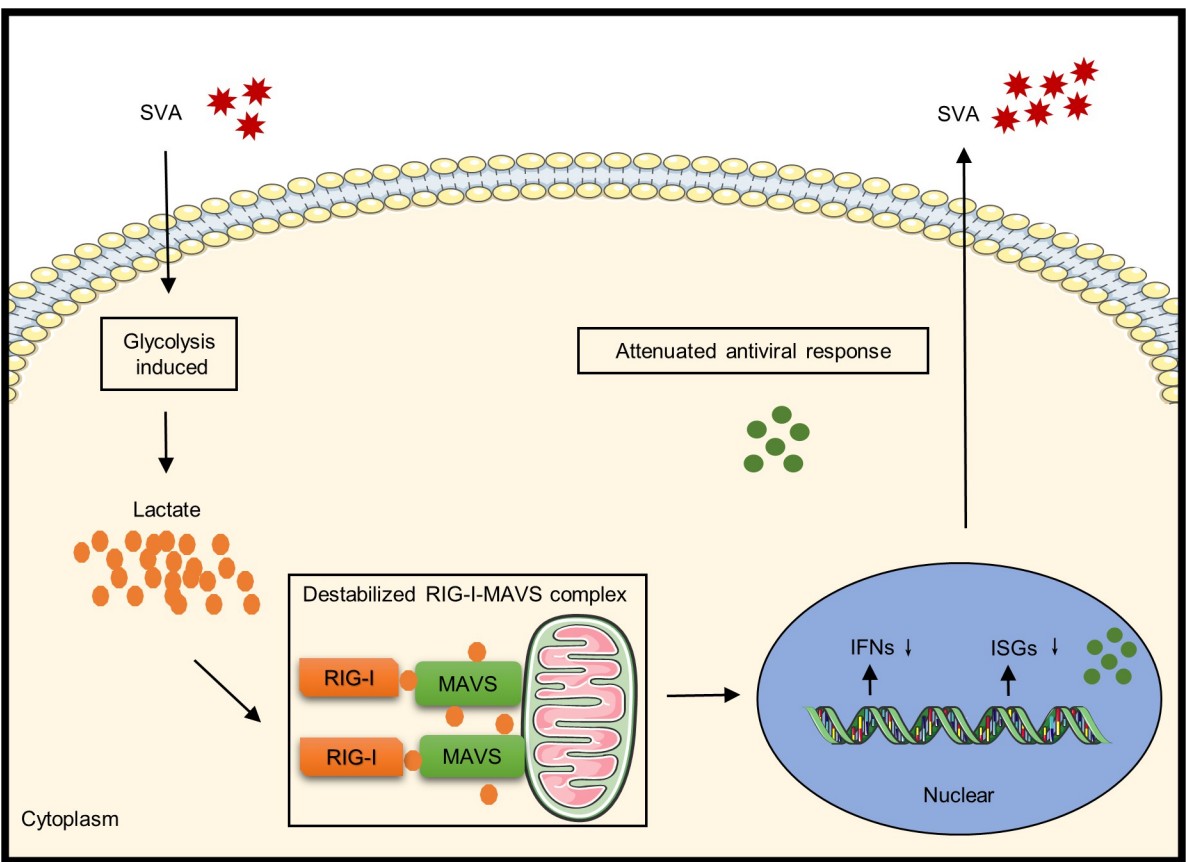

**Fig 9. Schematic overview of SVA-induced glycolysis that facilitates virus replication by promoting lactate production, which attenuates the interaction between MAVS and RIG-I.**

exploited by SVA for better replication in its host cells; therefore, regulation of glycolysis or directly targeting lactate production may be new targets against the virus.

## Materials and methods

### Ethics statement

Animal experiments were reviewed and approved by the Institutional Animal Care and Use Committee at South China Agricultural University (Approval Number: 2022F055) and were performed in accordance with the Animal Ethics Procedures and Guidelines of the People's Republic of China.

### Cells, virus, and reagents

Porcine kidney-15 (PK-15) and human embryonic kidney (HEK293T) cells were cultured at 37°C and 5% $CO_2$ in Dulbecco's modified Eagle's medium (DMEM; Gibco) supplemented with 10% fetal bovine serum (FBS; Gibco), 100 IU/mL penicillin, 100 g/mL streptomycin, and 2 mM L-glutamine (Gibco). The SVA strain, CH-GDFS-2018, was isolated by our group and propagated and titrated in PK-15 cells using the Reed–Muench method [46]. The SVA strain was incubated in a water bath at 70°C for 2 h to generate a heat-inactivated SVA (Heat-SVA).

Sodium oxamate, 2-deoxyglucose (2DG), sodium lactate, dichloroacetate (DCA), and glucose were purchased from Macklin (Shanghai, China). The 2DG (50 mM), sodium oxamate

(25 mM), DCA (5 mM), sodium lactate, and glucose were directly solubilized in DMEM or phosphate buffered saline (PBS) at the indicated concentrations. DMEM, PBS, and FBS were obtained from Gibco (Grand Island, NY, USA). Anti-HIF-1α, Anti-PGK1, anti-PKM, anti-GAPDH, anti-RIG-I and anti-β-actin primary antibodies were purchased from Santa Cruz (Shanghai, China). A primary antibody of rabbit anti-VP2 of SVA was prepared and stored in our laboratory [27]. The respective horseradish peroxidase (HRP)-conjugated secondary antibodies were obtained from Abcam (Cambridge, UK).

## Plasmid constructions and cell transfection

The swine PGK1, PKM, HIF-1α, and PDK3 coding sequences were obtained and subcloned into the pEGFP-N1 vector to generate pEGFP-PGK1, pEGFP-PKM, pEGFP-HIF-1α, and pEGFP-PDK3. The MAVS and RIG-I coding sequences were obtained and subcloned into the p3×Flag-CMV-10 or pCDNA3.1–3×HA vector to generate Flag-RIG-I and HA-MAVS. The constructed plasmids were confirmed by sequencing and transfected into PK-15 cells using the Lipo2000 transfection reagent (Invitrogen, USA) according to the manufacturer's instructions. At 24 h after transfection, the cells were infected with SVA, as described below.

## RNA extraction and quantitative real-time PCR

Total RNA was extracted using TRIzol reagent (Takara, Japan) and reverse transcribed into cDNA using a Reverse Transcription Kit (Takara, Japan), according to the manufacturer's instructions. qRT-PCR was performed using the SYBR Premix Ex Taq Kit (Takara, Japan) with specific primers (S1 Table) in a Light Cycler 480 (Roche, Switzerland). The relative expressions of the target genes were calculated using the $2^{-\Delta\Delta Ct}$ method, as described previously [47].

## Western blot

Cellular proteins were extracted using RIPA Lysis Buffer containing a protease inhibitor (DGCS Biotechnology, China), and the protein content was determined using the Bicinchoninic acid (BCA) Protein Assay Kit (Beyotime, China), according to the manufacturer's instructions. The same amounts of proteins were added to 5×SDS-PAGE loading buffer and denatured at 100˚C for 6 min. The proteins were then separated by 10% sodium dodecyl sulfate polyacrylamide gel electrophoresis (SDS-PAGE) and transferred onto nitrocellulose membranes at 150 V for 1 h (Bio-Rad, USA). After washing 3 times with Tris-buffered saline containing Tween-20 (TBST), the nitrocellulose membranes were blocked with 5% (w/v) skimmed milk at room temperature for 1 h and then incubated overnight with the indicated primary antibodies at 4˚C. The corresponding secondary antibodies were added and incubated for 1 h at room temperature. The proteins in the membranes were detected using an enhanced chemiluminescence reagent (Genview, USA).

## Virus titration

PK-15 cells were seeded in six-well plates at $2.5×10^5$ cells/well, and transfected with plasmids for 24 h, then infected with SVA at the indicated MOI. The PK-15 cell suspension was collected at the indicated time points and viral titration was performed. The 50% tissue culture infective dose ($TCID_{50}$) was calculated using the Reed–Muench formula [46]. In brief, PK-15 cells were maintained in 96-well cell culture plates, and virus samples to be titrated were prepared in a 10-fold serial dilution (from $10^{-1}$ to $10^{-8}$). The cells were infected by adding 100 μL per well of viral suspension, and after adsorption for 2 h, the inoculum was removed, and the cells were washed twice with DMEM. The cells were cultured in DMEM supplemented with 2% FBS for 7

days before the plates were read manually to determine the cytopathogenic effect (CPE). Cells were considered to show CPE when a loss of contact was observed between the cells.

## Measurement of ATP and lactate levels

The ATP levels in PK-15 cells and mouse serum were measured using an ATP assay kit (Beyotime, China), according to the manufacturer's instructions. The cells were lysed in ATP assay buffer, and ATP working solution was added and left to sit for 10 s at room temperature. The relative light units (RLUs) were then measured using a luminometer. The ATP levels were calculated based on a standard curve, and the sample concentrations were determined with a BCA kit (Beyotime, China). All samples and standards were measured in triplicate. The lactate levels in PK-15 cells were determined using lactate assay kits (Keming, China) according to the manufacturer's instructions. Samples and lactate standards were added in duplicate to 96-well plates. The pre-reaction mixture was added to each well and incubated at 37°C for 30 min. The absorbance of each well was then measured using a spectrophotometer at 530 nm to determine the lactate level based on the standard curve.

## Generation of RIG-I knockout PK-15 cells by CRISPR-Cas9- mediated genome editing

The sequence of 5'-CACCGAAACAACAAGGGCCCGACAG-3' for generating sgRNA targeting RIG-I was inserted and cloned into lentiCRISPR-V2 plasmid (Addgene, 52961) and transfected into HEK293T cells with the lentiviral packaging plasmids pMD2.G (Addgene plasmid #12259) and psPAX2 (Addgene plasmid #12260). Medium was changed 6 h post transfection. Forty-eight hours post transfection the virus containing medium was harvested and filtered through a 0.45 μm PDVF syringe filter (Millipore) to remove cell debris. PK-15 cells ($1.0 \times 10^6$) to be transduced were plated in an 8 cm$^2$ dish the day before transduction. After lentivirus added to the cells 48 h, cells were selected with 2.5 μg/mL of puromycin (Sigma Aldrich) medium for 4–6 days. Candidate knockout clones were screened by western blot with anti-RIG-I antibody and the positive named as RIG-I-KO PK-15 cells.

## Co-Immunoprecipitation

Cells transfected with relative plasmids were incubated in lysis buffer on ice, and the lysates were centrifuged at 12,000×g for 10 min. HA-Trap agarose beads were added to the supernatant and incubated for 1 h. After a rigorous washing with washing buffer, the beads were boiled with 2×SDS loading buffer, and the released proteins were analyzed by western blot.

## Animal experiments

Thirty-five 3-week-old female BALB/c mice were obtained from the Southern Medical University Laboratory Animal Center. All mice were maintained in pathogen free barrier facilities and randomly assigned to 7 groups (negative control, SVA-high glucose, SVA-low glucose, SVA-oxamate, SVA-lactate, SVA-oxamate + lactate, and SVA-challenged group), with 5 mice in each group. The mice were infected with SVA at $2 \times 10^{-7}$ TCID$_{50}$ based on the mouse infection model described by Li et al. [29] and were treated with different reagents dissolved in PBS at 5 d.p.i. and 6 d.p.i. At 7 d.p.i, whole blood was collected, and the serum was separated. The mice were then euthanized, and tissues (heart, liver, spleen, kidney, and duodenum) were collected from all groups and stored. A part of the tissue samples were fixed in 4% paraformaldehyde and embedded in paraffin, cut into 4 μm thick sections, then used hematoxylin and eosin (HE).

## Statistical analysis

The data are expressed as the mean ± standard deviation from at least three independent experiments. Statistical analysis was performed using Student's $t$-test and one-way analysis of variance (ANOVA) using GraphPad Prism version 8.0 survey software (GraphPad Software, La Jolla, CA). A value of $P < 0.05$ was considered statistically significant.

## Supporting information

**S1 Fig. SVA infection induces glycolysis in HEK293T cells. (A, B)** HEK293T cells were infected with SVA at an MOI of 1. Cell lysates were collected at 48 h.p.i. and intracellular ATP levels and the lactate were measured. **(C)** Cells were infected with SVA at an MOI of 1 and lysed in RIPA buffer at 48 h.p.i. The expression levels of PGK1, PKM, and VP2 were analyzed by western blot. **(D)** The grayscale analysis of PGK1, PKM, and VP2 protein. All data represent the means ± SD (Student's t test) (*P < 0.05, **P < 0.01, ns, not significant).
(TIF)

**S2 Fig. Glycolysis induced by CH-GDQC-2017 and CH-GDZS-2019.**
(TIF)

**S1 Data. Excel spreadsheet containing, in separate sheets, the numerical data and statistical analysis for Figure panels 1A, 1B, 1C, 1E, 2A, 2B, 2C, 2D, 2E, 2F, 3B, 3D, 3E, 3G, 4A, 4B, 4C, 4D, 4E, 4F, 4G, 4H, 4I, 4J, 4K, 4L, 5A, 4B, 5C, 5E, 5F, 5G, 5H, 5I, 5J, 5K, 5L, 6D, 6E, 6F, 6G, 7B, 7C, 7E, 7F, 7G, 7H, 7I, 7J, 7K, 7L, 7M, 7N, 7O, 7P, 7Q, S1A, S1B, S1D, and S2.**
(XLS)

**S1 Table. Primer sequences for qRT-PCR.**
(DOC)

## Author Contributions

**Conceptualization:** Huizi Li, Wenbao Qi, Zhangyong Ning.

**Data curation:** Huizi Li, Cunhao Lin.

**Formal analysis:** Huizi Li.

**Funding acquisition:** Zhangyong Ning.

**Investigation:** Huizi Li, Cunhao Lin, Zhenzhen Sun, Zhenxin Xie.

**Methodology:** Huizi Li, Zhenzhen Sun, Zhenxin Xie.

**Project administration:** Zhangyong Ning.

**Visualization:** Cunhao Lin, Zhenzhen Sun.

**Writing – original draft:** Huizi Li, Zhangyong Ning.

**Writing – review & editing:** Huizi Li, Wenbao Qi, Weixin Jia, Zhangyong Ning.

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
