## [Decision Letter · Decision Letter 0]

27 Jan 2023

Dear Dr. Ning,

Thank you very much for submitting your manuscript "Senecavirus A-induced glycolysis facilitates virus replication by promoting lactate production that attenuates the interaction between MAVS and RIG-I" for consideration at PLOS Pathogens. As with all papers reviewed by the journal, your manuscript was reviewed by members of the editorial board and by several independent reviewers, who found your study interesting. In light of the reviews (below this email), we would like to invite the resubmission of a significantly-revised version that takes into account the reviewers' and editor's comments.

- We encourage you to corroborate the main cell culture findings (Fig 5D-F) with TCDID50 or plaque assay. From the RNA and protein levels the reader can clearly see the trends but not the effect size. To which extent was the replication inhibited?

- Does glycolysis facilitate senecavirus reproduction solely via RIG-I signaling or other pathways involved as well? To this end, an experiment with infection in RIG-I knockdown (e.g., shRNA or CRISPRi) plus/minus glycolysis inhibitors plus/minus lactate would be very informative.

- More information on tissue pathology in mice is required, as suggested by Reviewer #2.

- Lines 251-254: Please correct name superscripts of virus proteins, e.g. 2B, 2C etc, (see PMID: 28884666  for guidance and links).

We cannot make any decision about publication until we have seen the revised manuscript and your response to the reviewers' comments. Your revised manuscript is also likely to be sent to reviewers for further evaluation.

Sincerely,

Peter V Lidsky

Guest Editor

PLOS Pathogens

Alexander Gorbalenya

Section Editor

PLOS Pathogens

Kasturi Haldar

Editor-in-Chief

PLOS Pathogens

orcid.org/0000-0001-5065-158X

Michael Malim

Editor-in-Chief

PLOS Pathogens

orcid.org/0000-0002-7699-2064

Reviewer's Responses to Questions

**Part I - Summary**

Reviewer #1: This manuscript reports investigation of effect of glucose metabolism on the SVA infection. The authors performed in vitro and in vivo characterization of glycolysis related pathway in SVA infection. They showed that SVA infection can induce glycolysis and production of lactate, which can inhibit RLR signaling. Overexpression of glycolytic enzymes promoted SVA replication. The findings are very interesting.

Reviewer #2: In the manuscript "Senecavirus A-induced glycolysis facilitates virus replication by promoting lactate production that attenuates the interaction between MAVS and RIG-I", the authors evaluated the new mechanism used by SVA to enhance its replication from the perspective on the relationship between metabolism and innate immunity of the host. The results also suggest glycolysis or lactate may be new targets against this swine emerging virus. The work is well designed and the manuscript is clearly written and structured.

Reviewer #3: Senecavirus causes porcine idiopathic vesicular disease, which is harmful to the pig industry, and forms persistent infection and weaken the immune system of the pig. This manuscript focused on the effect of Senecavirus on innate immunity, i.e., IFN production. The novel idea was started from the aspect that viral infection affected cellular sugar metabolism, which then affected IFN production, and a very positive result was obtained. Therefore, the manuscript provided a new idea for controlling the disease from the aspect of metabolic control.

**Part II – Major Issues: Key Experiments Required for Acceptance**

Reviewer #1: Its clinical value to control SVA in pigs remains further confirmed.

Are the findings of this study limited to the single strain used or to all SVA strains? Is there any unique changes in the CH-GDFS-2018 strain? This needs to be discussed.

Reviewer #2: 1. Cell lines commonly used for SVA include PK-15 and HEK293T cells. Whether SVA infection-mediated glycolysis exists in non-porcine origin cells, HEK293T? It is better to clearly state.

2. In Figure 8, schematic overview, this figure has clearly defined the whole research context. It is recommended to remove the dotted line outside the cell membrane.

3. In animal experiments, the results showed the difference of virus load in tissues collected with various treatment in different groups. Readers would like to know whether this change of viral load caused by different treatment lead to pathological or histopathological change? This is related to the effect of different factors on the pathogenicity of the virus.

Reviewer #3: 1. It is mentioned that glycolysis promotes viral replication, but for viral replication, the authors measures the expression of SVA VP2 protein. It suggests that viral titers should be preferably determined. Otherwise, explain the feasibility of VP2 protein test.

2. In Figure 1, differences in mRNA levels of some proteins related to glycolysis such as HK2, PFKM, PKM, PGK1 and HIF-1α were detected after SVA virus infection, but only PGK1 and PKM were detected during protein level detection. Please explain why the latter two were selected for protein level detection

**Part III – Minor Issues: Editorial and Data Presentation Modifications**

Reviewer #1: Page 2, abbreviations of many molecules and pathways needed spell out.

The manuscript needs to be improved since several places lack of scientific writing. Lines 241-243, what is the meaning of “this first outbreak”?

Reviewer #2: 1. Abstract Page 2, line 35 : Please specify what experiments while not “Further”;

2. Page 7, lines: 141-145. Please note that contents in brackets for 2DG, oxamate and DCA was information belongs to the methods;

3. Page 25, lines: 605. “proteins” should be “detected proteins”.

Reviewer #3: FIG. 1C, the ordinate icon "Relative viral RNA" should be modified to "Relative expression levels of mRNA".

PLOS authors have the option to publish the peer review history of their article (what does this mean?). If published, this will include your full peer review and any attached files.

Reviewer #1: No

Reviewer #2: **Yes: **Yingjun Lv

Reviewer #3: No
---

## [Decision Letter · Decision Letter 1]

12 Apr 2023

Dear Dr. ning,

Thank you very much for submitting your manuscript "Senecavirus A-induced glycolysis facilitates virus replication by promoting lactate production that attenuates the interaction between MAVS and RIG-I" for consideration at PLOS Pathogens. As with all papers reviewed by the journal, your manuscript was reviewed by members of the editorial board and by several independent reviewers. The reviewers appreciated the attention to an important topic. Based on the reviews, we are likely to accept this manuscript for publication, providing that you modify the manuscript according to the recommendations below and Reviewer #3 request:

Please fix the protein names that are misrepresented: Lines 251/254: “2B proteinase (2Bpro), 2C proteinase (2Cpro), 3C proteinase (3Cpro), 3D proteinase (3Dpro)”.

Proteinase = protease, it means a specific enzymatic activity, ability of the protein to cleave other proteins. For all picornaviruses, this activity was ascribed only to 3C protein, that can have "pro" superscript: 3Cpro. Likewise, 2C is a helicase and 3D is a polymerase, they should be spelled as 2Chel and 3Dpol, respectively. 2B does not have enzymatic activity and should be spelled without a superscript. Please see this paper for guidance doi:10.1099/jgv.0.000911 .

Sincerely,

Peter V Lidsky

Guest Editor

PLOS Pathogens

Alexander Gorbalenya

Section Editor

PLOS Pathogens

Kasturi Haldar

Editor-in-Chief

PLOS Pathogens

orcid.org/0000-0001-5065-158X

Michael Malim

Editor-in-Chief

PLOS Pathogens

orcid.org/0000-0002-7699-2064

Dear Dr. Ning,

We are up to accept the manuscript in its present form. No additional experiments are required.

Please fix the protein names that are misrepresented: Lines 251/254: “2B proteinase (2Bpro), 2C proteinase (2Cpro), 3C proteinase (3Cpro), 3D proteinase (3Dpro)”.

Proteinase = protease, it means a specific enzymatic activity, ability of the protein to cleave other proteins. For all picornaviruses, this activity was ascribed only to 3C protein, that can have "pro" superscript: 3Cpro. Likewise, 2C is a helicase and 3D is a polymerase, they should be spelled as 2Chel and 3Dpol, respectively. 2B does not have enzymatic activity and should be spelled without a superscript. Please see this paper for guidance doi:10.1099/jgv.0.000911 .

With regards,

Peter Lidsky

Reviewer Comments (if any, and for reference):

Reviewer's Responses to Questions

**Part I - Summary**

Reviewer #1: my comments were addressed

Reviewer #2: The authors answered all my questions, I have no questions at present.

Reviewer #3: The article has been modified according to the questions raised by the reviewers. The data is completed and the article has been well improved.

**Part II – Major Issues: Key Experiments Required for Acceptance**

Reviewer #1: (No Response)

Reviewer #2: I think no other experiments are needed.

Reviewer #3: None

**Part III – Minor Issues: Editorial and Data Presentation Modifications**

Reviewer #1: (No Response)

Reviewer #2: No

Reviewer #3: In Figure 3 E and F. For consistency, it is suggested to place HG column on the left and LG column on the right.

PLOS authors have the option to publish the peer review history of their article (what does this mean?). If published, this will include your full peer review and any attached files.

Reviewer #1: No

Reviewer #2: No

Reviewer #3: No

Figure Files:

Data Requirements:

Reproducibility:

References:

---

## [Editor Report · Decision Letter 2]

18 Apr 2023

Dear Dr. ning,

We are pleased to inform you that your manuscript 'Senecavirus A-induced glycolysis facilitates virus replication by promoting lactate production that attenuates the interaction between MAVS and RIG-I' has been provisionally accepted for publication in PLOS Pathogens.

Best regards,

Peter V Lidsky

Guest Editor

PLOS Pathogens

Alexander Gorbalenya

Section Editor

PLOS Pathogens

Kasturi Haldar

Editor-in-Chief

PLOS Pathogens

orcid.org/0000-0001-5065-158X

Michael Malim

Editor-in-Chief

PLOS Pathogens

orcid.org/0000-0002-7699-2064
---

## [Editor Report · Acceptance letter]

26 Apr 2023

Dear Dr. Ning,

We are delighted to inform you that your manuscript, "Senecavirus A-induced glycolysis facilitates virus replication by promoting lactate production that attenuates the interaction between MAVS and RIG-I," has been formally accepted for publication in PLOS Pathogens.

Best regards,

Kasturi Haldar

Editor-in-Chief

PLOS Pathogens

orcid.org/0000-0001-5065-158X

Michael Malim

Editor-in-Chief

PLOS Pathogens

orcid.org/0000-0002-7699-2064